# Development and Validation of the Chinese Frailty Screening Scale: A Study among Community-Dwelling Older Adults in Shanghai

**DOI:** 10.3390/ijerph191811811

**Published:** 2022-09-19

**Authors:** Bo Ye, Yi Wang, Hao Chen, Yingwei Chen, Huihui Yan, Hua Fu, Zhijun Bao, Junling Gao

**Affiliations:** 1Huadong Hospital Affiliated to Fudan University, Shanghai 200040, China; 2School of Public Health, Fudan University, Shanghai 200032, China; 3Health Communication Institute, Fudan University, Shanghai 200032, China; 4Shanghai Key Laboratory of Clinical Geriatric Medicine, Huadong Hospital, Shanghai 200040, China; 5Collaborative Innovation Cooperative Unit, National Clinical Research Center for Geriatric Diseases, Shanghai 200032, China; 6Core Unit, Shanghai Clinical Research Center for Geriatric Diseases, Shanghai 200032, China

**Keywords:** frailty, assessment tool, disability, older adults, healthy aging

## Abstract

Background: Based on intrinsic capacity (IC) as defined by the World Health Organization, an accelerated decline may be an important precursor of frailty among older adults; however, there is a lack of validated instruments that both screen for frailty and monitor IC. This study aims to develop a comprehensive and acculturative frailty screening scale to determine healthy aging among older Chinese adults. Setting and participants: A cross-sectional and a cohort study both based on community-dwelling older adults aged 65 and older. Methods: This study mainly consisted of two parts. First, the selection and revision of 20 items related to frailty based on a literature review, expert consultation, and stakeholder analysis; second, a cross-sectional study was conducted to simplify the scale and test the reliability and validity of the new frailty screening tool. The fatigue, resistance, ambulation, illness, and loss of weight (FRAIL) scale, the Tilburg frailty indictor (TFI), and a 49-item Frailty Index (FI) were investigated as criteria. Additionally, a cohort study in Shanghai was conducted to verify the predictive validity of the new screening scale. The disability measured by the activity of daily living (ADL), instrumental activity of daily living (IADL) and all-cause mortality were documented as outcomes. Results: A 10-item Chinese frailty screening scale (CFSS-10) was successfully developed and validated. It presented a Cronbach’s α of 0.63 and an intraclass correlation coefficient of 0.73, which indicated good reliability. Taking the other frailty tools as criteria, Kappa values of 0.54–0.58 and an area under the curve of 0.87–0.91 showed good validity. The results of the log-binomial and Poisson models showed a high score, which predicted a higher risk of disability and all-cause mortality. An optimal cut-off point of 5 gave an excellent prediction of one-year disability. Conclusions: The CFSS-10 has good validity and reliability as a quick and acculturative frailty screening scale for community-dwelling older adults in Shanghai. It may also supplement existing frailty screening tools.

## 1. Introduction

Frailty is a progressive decline of physiological reserves that increases vulnerability of older adults to stressors and adverse health outcomes, including disability, falls, hospital admissions, and mortality [1,2]. A meta-analysis showed the prevalence of frailty among adults aged ≥50 was 12–24% worldwide [3]. Its prevalence ranged from 6–25% for peoples aged 65–74 and ≥85 with an average of 10% in China [4]. However, the global or regional prevalence of frailty is not yet accurately known, and it has different operational definitions. Even so, frailty represents a public health priority for its increasing prevalence [5].

To cope with increasing aging populations, the World Health Organization (WHO) proposed a framework of healthy aging [6] and emphasized the importance of developing an approach to screen for decreasing intrinsic capacity (IC), defined as “the combination of the individual’s physical and mental capacities” [7]. It was developed based on the evidence of frailty built over decades and nested into the framework of healthy aging [8]. “Integrated Care for Older People” (ICOPE) was developed as a guide for screening for the loss of IC in six areas: cognitive decline, mobility, malnutrition, vision impairment, hearing loss, and depressive symptoms [9]. The IC indicators, such as walking speed, grip strength, and exhaustion, were also used in frailty definitions, while the cognition tests were also used to screen for cognitive frailty [10]. With the exception of cognitive and sensory capacity, the other capacities overlap with frailty phenotypes. In fact, the frailty index model, also named cumulative deficit model [11], and the multidimension frailty model [12] covered the six aspects, with the former including physical function, multimorbidity, cognition, and psychosocial factors, and the latter comprising physical, psychological, and social domains of frailty. Recently, the ICOPE tool was also recommended as a frailty screen by the International Conference of Frailty and Sarcopenia Research (ICFSR) consensus guidelines [13]. The IC could be used throughout one’s life and regarded as a comprehensive aspect of an individual’s functional capabilities, while frailty was regarded as an accelerated decline before the occurrence of disability in later life. To a large extent, a decrease in IC is likely an important precursor cause for either prefrailty or frailty [10,13]. According to both existing assessment frameworks, an abnormal decline in IC may coincide with the onset of frailty. Using a life-course approach might increase our understanding of how frailty and its risk factors develop in earlier life stages and contribute to the development of public health strategies for frailty prevention [14].

In the past decade, there have been numerous tools developed to screen or assess frailty [15]. However, there is still no standard assessment instrument [14]. The most widely used measures are the Fried frailty phenotype (FP) [2] and the frailty index (FI) [11], but these are not easily implemented in a large population study or busy clinics: the former requires objective measurements and the latter needs to collect too much information [15]. In recent years, a Chinese self-reported frailty screening questionnaire (FSQ) was developed based on modified Fried FP criteria [16] and validated at different settings [17]. As we know, the FP model only assesses the physical aspect of frailty. In addition, previous studies developed scales based on the FI for older adults in China [18,19]. One of them, the FI-35 with an oversize of 11 dimensions, might produce a potential third-order model [20]. Moreover, it was not easy to screen frailty in a large-scale population, which was the same disadvantage with other FI-based instruments [15]. Subsequently, several simple and effective screening questionnaires emerged. The FRAIL scale, a simple tool to screen for frailty, has been used among Chinese older populations [21,22]. However, its weakness is that it eliminates low physical activity, which may change the theoretical model of the FP [23] and it is the same as the FP in that it lacks the mental aspect of frailty. Frailty is increasingly considered a multidimensional concept, comprising physical, mental (psychological and cognitive), and social features. A Chinese version of the Tilburg frailty indictor (TFI) was also developed to measure frailty among community-dwelling older adults [24]. Whereas, social frailty is similar to social isolation, which was considered as the external cause of frailty [25]. Most recently, a Japan frailty scale (JFS) was developed based on the aging concept of Kampo medicine; this was a new attempt in assessing frailty [26]. Similarly, the significant association of traditional Chinese medicine (TCM) constitution and frailty status was also observed in Chinese older adults [27]. On the whole, physical frailty and cognitive frailty are widely accepted concepts, which was probably the reason why the WHO compared IC with frailty [28] and why researchers regard them as related concepts [8,10]. Although the physical and mental aspects of the TFI covered the six areas of the IC, it was only a subscale of the model and lacked relatively independent reliability and validity analyses and cultural adaptability tests.

In the current context of China, there is lack of a locally developed, multi-dimensional, and brief frailty screening tool. Importantly, based on the close relationship between frailty and IC, a frailty tool combined with the ICOPE framework could be used for rapid frailty screening and IC monitoring. This as well as fits the consensus that IC monitoring in the early stage of aging is beneficial for identifying the process of frailty. Therefore, this study aims to develop and validate a Chinese frailty screening scale based on both the definition of frailty (physical and cognitive aspect) and ICOPE framework that rapidly detects IC and screens frailty in older Chinese adults.

## 2. Methods

The study was approved by the Institutional Review Board of School of Public Health Fudan University, and all participants provided written informed consent. It consisted of two parts:

### 2.1. Part One: Drafting the Chinese Frailty Screening Scale (CFSS)

The flow chart is shown in Figure 1. First, a research group collected and discussed existing frailty tools, overseen by Azzopardi et al. [29], and recommended by the Asia-Pacific Clinical Practice Guidelines for the Management for Frailty [22]. We reviewed 66 frailty instruments and constructed a frailty scale with 33 items (Appendix A) based on the framework of FP and ICOPE after eliminating duplicate or similar items from January to March 2019.

Second, we invited 27 experts (average age 55.48; standard deviation (SD) = 8.68) specializing in gerontology, public health, nursing, and psychology from nine cities in China to participate in a two-round Delphi method of expert consultation [30] from March to June 2019. In the first round of 27 advisory emails, 25 were returned for a rate of 92.6%. The second round was the same as the first. In the feedback emails, the experts rated each item on a 5-point Likert scale: from 1 (not very important) to 5 (very important) and provided detailed advice. Moreover, we invited 20 experts for a seminar to discuss the content of the frailty scale on 23 September 2019. Based on the rating score and expert discussions, a 20-item CFSS (CFSS-20) was obtained (Appendix A), with 12 items retained, 4 items revised, 4 items replaced, and 14 items dropped.

Furthermore, we determined if the language of the items were suitable. We invited 10 older adults to participate in a pilot survey, which showed that the items were easy to understand.

### 2.2. Part Two: Revising and Validating the CFSS

First, a cross-sectional study among community-dwelling older adults was conducted to simplify the items of the CFSS-20. Multi-stage random sampling was used to select subjects. Five cities were purposefully selected from five geographic areas: east, west, south, north, and center. Two communities, an urban and suburban/village in each city, were then randomly selected. Simple random sampling was used to select older adults living in the communities. Samples were also added from other cities because of the influence of the SARS-CoV-2 pandemic. A total of 1083 participants aged 65 and older were recruited from November 2019 to May 2020. After eliminating participants with missing data, the valid response rate was 98.1% (1062/1083) (sample 1). To determine the test–retest reliability of the simplified frailty screening scale, 89 participants were assessed twice using the simplified frailty screening scale within a 2-week interval.

In this cross-sectional study, the sample size was calculated by the formula: *n* = Z^2^ × (1 − P)/(P × ε^2^). The indicator Z is the percentile corresponding to an area of 0.975 (1 − 0.05/2) under the standard normal distribution and the prevalence of frailty was 10–15%. The indicator of P was set to 12.5% and ε to 0.2. The minimum sample size was 673, but an additional 20% was added in consideration of the response rate. Therefore, it was better to enroll more than 808 participants. The participants received face-to-face interviews from trained community health workers to complete the questionnaire. Those unable to communicate effectively or with severe cognitive impairment were excluded because of the self-reporting nature of data.

Second, a cohort study based on community-dwelling older adults in Shanghai was conducted to verify the predictive validity of the simplified frailty screening scale. Multi-stage random sampling was also used to select participants. Seven communities were randomly selected from the Songjiang district. Simple random sampling was used to select 300 families with older adults aged 65 years and older in each community through a household registration information system. A total of 2260 participants were originally recruited from June to December 2020 for an investigation at baseline. After eliminating 111 for invalid questionnaires or missing data, the valid response rate was 95.1% (2149/2260). During the next year, we followed the cohort from July to December 2021. One hundred and forty-one participants were lost to follow-up and 37 died, with the death information obtained from family members or community workers during follow-up, thus the analytic sample was 2008 (93.4%) (Sample 2).

## 3. Measurements

To revise and validate the CFSS-20, the following measurements were used:(1)The fatigue, resistance, ambulation, illness, and loss of weight (FRAIL) scale was derived from a consensus of a European, Canadian, and American geriatric advisory panel [31]. It is a quick and valid tool applied globally [32] and had been used and validated among Chinese older populations [21]. It consists of 5 simple questions and ranges from 0 to 5 with 1 point for each question (0 = best; 5 = worst): The scores were frail (3–5), prefrail (1–2), and robust (0).(2)The TFI, a self-reported 15-item questionnaire, addressed three domains. The physical domain consisted of 8 items (e.g., difficulty in walking or hearing) ranging from 0 to 8 points. The psychological domain had 4 items (e.g., depressive symptoms, coping) ranging from 0 to 4 points. The social domain had 3 items (e.g., living alone, social support) ranging from 0 to 3 points. A high score indicates more frailty, and a total score of 5 or more was regarded as frail [33]. The Chinese version of TFI has been validated in China [24].(3)The FI, which was constructed using a standardized procedure [34] included 49 self-reported items referring to the list of FI constructed by previous studies [35,36]. A summary score of 0–49 with one point per item was used to construct a total score/49. It consisted of the self-reported presence of diseases (20 items: e.g., hypertension, diabetes mellitus, coronary heart disease); geriatric symptoms (12 items: e.g., vision impairment, hearing loss, falls in the previous year); difficulties in performing basic and instrumental activities of daily living (14 items: e.g., dressing, transferring, shopping); cognitive decline (1 item) using a Chinese version of the mini-mental state examination (MMSE) [37] total score < 27; depression (1 item) used a Chinese version of the geriatric depression scale (GDS-15) [38] for a total score ≥ 8; and self-rated poor health (1 item). An FI score more than 0.25 was classified as frail.(4)Physical function was evaluated by Katz as the activity of daily living (ADL) and by Lawton as the instrumental activity of daily living (IADL). Katz’s ADL consisted of feeding, continence, transferring, toileting, dressing, and bathing [39]. Lawton’s IADL comprises using the telephone, shopping, preparing food, housekeeping, doing laundry, using transportation, handling medications, and handling finances [40]. For each activity, participants were asked if they had no difficulty, had difficulty, or were unable to perform the tasks from 0 to 2 points. The presence of ADL or IADL deficits to any positive individual score (i.e., equivalent to more than ‘‘0”) indicated disability [41]. Body mass index (BMI, the ratio of weight and squared height, kg/m^2^) was also included.

Demographic characteristics included age, sex, education (illiteracy, primary school, middle school, high school, or college and above), and marital status (never married, married, widowed, or divorced).

## 4. Statistical Analysis

The data was presented as mean ± standard deviation (SD) or number and percentage (%). The differences in distribution between the two groups were evaluated by independent Student’s *t* tests for continuous variables or Chi-square tests for categorical variables. To achieve rapid screening in population, binary logistic regression with a stepwise method was used to streamline items of the CFSS-20 as independent variables and functional disability as the dependent variable. Considering the effect of the excluded items on the frailty model, all the 10 and 20 items were included in the logistic model for fitting (Appendix A).

The test–retest reliability of the simplified frailty screening scale was assessed using the intraclass correlation coefficients (ICC), where less than 0.40 indicated poor consistency and more than 0.70 showed good reliability [42]. Spearman’s correlation and factor analysis were performed to evaluate the convergent and divergent validities, respectively. The area under the curve of the receiver operating characteristic (AUC–ROC) was also calculated to set the optimal cutoff point and evaluate criteria validity [43]. Kappa values were calculated for the agreement of the simplified frailty screening scale with the existed frailty tools including FRAIL, TFI, and FI. A value above 0.40 indicated moderate agreement and above 0.60 indicated strong agreement [44]. The log-binomial and Poisson models were used to examine the predictive validity of the simplified frailty screening scale.

All statistical analyses were performed using SPSS 22.0 (SPSS Inc., Chicago, IL, USA). All statistical tests were two-tailed, and *p* < 0.05 was considered statistically significant.

## 5. Results

### 5.1. Participant Characteristics

The characteristics of sample 1 and sample 2 are shown in Table 1. One thousand and sixty-two and 2008 older adults with a mean age of (76.7 ± 7.2) and (72.4 ± 6.1) years, and 47.0% and 46.7% of men were included in sample 1 and sample 2, respectively. In sample 1, the prevalence of frailty assessed by the three frailty questionnaires were 23.7% (252/1062, FRAIL), 31.9% (237/743, TFI), and 18.6% (74/397, FI), respectively. Older adults with one or more difficulties in ADLs or IADLs were 499 (47.0%) in sample 1, and the numbers were 459 (22.8%) at baseline and 374 (24.1%) after one-year follow-up in sample 2.

### 5.2. Simplified CFSS

Logistic regression indicated that eight items of the CFSS-20 were significantly associated with functional disability (Appendix A). Taking account of the ICOPE framework, we recuperated two items that presented sensory capacity, which were two of the important IC components. Thus, we developed three versions of CFSS, the CFSS-20, the CFSS-8, and the CFSS-10 (Table 2). The distributions of the CFSS-20, the CFSS-8, and the CFSS-10 score were shown in Appendix A.

### 5.3. Reliability

The Cronbach’s alpha coefficient showed the internal consistency reliability of CFSS-10 was 0.63, and the test–retest reliability was also good with an ICC of 0.73 in sample 1. The coefficient was 0.61 in sample 2.

### 5.4. Construct Validity

As shown in Table 3, the convergent validity of the CFSS-10 evaluated by the Spearman rank correlation coefficients ranged from 0.28 to 0.59 between each item score and the total scale score and were statistically significant. The divergent validity was also good according to the factor analysis. Four factors were extracted that explained 58.0% of the variance. All ten items demonstrated moderate-to-strong loading (>0.50). Factor 1, representing mental function (i.e., cognition, depression), explained 24.8% of the variance; factor 2, representing physical function (i.e., illnesses, malnutrition), explained 11.8%; factor 3, representing sensory function, explained 11.2%; and factor 4, representing mobility, explained 10.1%.

### 5.5. Criteria Validity

Table 4 presents the criteria validity and diagnostic accuracy of the CFSS-10 from a ROC curve analysis. Prevalence of frailty by the cutoffs of ≥3, ≥4, and ≥5 for the CFSS-10 total score was 49.5 (526/1062), 34.6 (367/1062), and 21.5% (228/1062), respectively. Moderate agreement (kappa 0.46–0.58) was found between the CFSS-10 and the other frailty tools. Taking the FRAIL, TFI, and FI as criteria, the AUCs of the CFSS-10 were 0.911 (95% CI: 0.894–0.928), 0.874 (95% CI: 0.847–0.900), and 0.874 (95% CI: 0.826–0.921), respectively (Figure 2). The results of the AUC–ROC for the CFSS-20 and the CFSS-8 are shown in Appendix A.

### 5.6. Predictive Validity

Table 5 shows excellent predictive validity for a one-year disability and a positive total score association (r = 0.215, *p* < 0.001). The log-binomial models indicated a higher CFSS-10 score, which predicted a higher risk of one-year disability after adjusting for age, sex, education, marital status, and comorbidity. The univariate analyses showed that a higher score positively correlated with one-year all-cause mortality, while the trend was not significant in the multivariate analysis. The results of multivariable models excluded comorbidity variable are shown in Appendix A. The results of CFSS-8 predictions for one-year disability and all-cause mortality are shown in Appendix A, and the results of CFSS-8 and CFSS-10 predictions for ADL and IADL disability are separately presented in Appendix A. In addition, the cut-off point of 5 for the CFSS-10 showed a statistically significant prediction of one-year disability. Compared with 0 for the CFSS-10, the adjusted RRs for 1–4 and ≥5 points were 1.20 (95% CI: 0.99–1.46) and 1.57 (95% CI: 1.08–2.28), respectively (Figure 3a). Compared with the 0–4 points of the CFSS-10, the adjusted RR for ≥5 was 1.37 (95% CI: 0.98–1.92) (Figure 3b).

## 6. Discussion

Although some frailty screening instruments have been used in China, most of these scales were translated from other languages, which may make them culturally inaccurate. More recently, researchers from non-native English-speaking countries have developed their own localized frailty screening tools [26,45,46]. This study developed and validated a frailty screening scale in a Chinese context to screen older adults on the basis of the frailty definition and the ICOPE framework. Hoogendijk et al. [14] noted that it might increase understanding of how frailty develops in earlier life stages from a life-course perspective. Especially in early aging, the decline in IC is close to the frailty curve, which offers the best opportunity to find and manage potential frailty for older adults. The CFSS-10 was developed through a complete scale development process. The content of the CFSS-10 was revised from two rounds of expert native consultation and discussion, which provided sufficient knowledge and experience for the CFSS-10. At the beginning of this study, a pilot survey indicated that the language of the CFSS-10 was suitable and easy to understand for older Chinese adults.

This study proved the CFSS-10 to be a reliable instrument frailty screening tool and that it had an acceptable internal consistency reliability of 0.63 and a good test–retest reliability of 0.73. The components of the CFSS-10 cover the most important elements of frailty in different functional areas; thus, a measure without high internal consistency was not a problem [33]. Moreover, for our purposes, it was beneficial for the CFSS-10 to have a test-retest reliability higher than its internal consistency reliability. A good test–retest reliability represents favorable assessment stability, which was a more desirable feature for measuring the physical functions.

Good validity of the CFSS-10 was also shown by both the factor analysis and the moderate-to-strong coefficients of agreement with the alternative frailty tools. In the current study, four factors with eigenvalues greater than 1 were extracted, and all loadings on factors were greater than 0.50, both of which indicated good construct validity. Factor 1 (cognitive) corresponded to both the psychological and cognitive dimensions of frailty [12,47] and two aspects of ICOPE (depressive symptoms and cognitive decline) [9]. Factor 2 (physical) comprised core aspects of frailty (i.e., illness, exhaustion, and strength) [31] and malnutrition in ICOPE [9]. Factor 3 (sensory) was absent in some physical frailty scales but constituted two out of six important components of ICOPE [9]. It is worth mentioning that the sensory functions were regarded as frailty measures or markers [12,48,49]. Factor 4 (inactive) indicated low physical activity in the FP [41,50] and reduced mobility in ICOPE [9]. The ICOPE was developed with the expectation of measuring individual ICs, which is an essential approach to achieve healthy aging [9]. Monitoring the early loss of IC can identify people as prefrail or frail so that early intervention can begin [13].

Criteria validity was examined through diagnostic accuracy and the agreements between the test scale and external criteria tools. The FRAIL scale, the TFI, and the FI were used as the external criteria, which showed robust reliability for detecting frailty in older adults and for strongly predicting adverse outcomes [21,33,51,52,53,54]. Our results showed that the 4-point and 5-point CFSS-10 cutoffs both had moderate kappa coefficients (>0.50) as did the other frailty scales, which indicated good criteria validity. Appendix A shows the excellent criteria validity of the CFSS-10 compared with CFSS-8, and that the CFSS-10 had the same validity as the CFSS-20 with only half the items.

The correlations between the CFSS-10 score and the risk of one-year disability indicated excellent predictive validity for disability. However, the predictive ability of the CFSS-10 on one-year all-cause mortality was not significant for a higher score of the CFSS-10; this is probably because the short follow-up duration of the study meant extremely low incidences of all-cause mortality, which led to a higher possibility of the false negative type II error. On the other hand, those with a higher score of the CFSS-10 at baseline might get more attention and health care, so we did not observe the outcomes in a short period of follow-up. In fact, considering population prevalence and other relative risks, the effect of frailty on mortality could be not high [55]. Although numerous studies have shown significant association between frailty and mortality, the evidence for short-term effects of frailty on mortality was not enough at a community setting [56]. As the CFSS-10 score increased, it represented a higher risk of disability in daily activities, which was similar with previous studies [21,50,57]. The strong linear association between the CFSS-10 score and disability indicated it could be used to monitor physical functioning before disability. It also supported the consensus that frailty is largely a former stage of disability [31,32]. The log-binomial model suggested that the cut-off points of 1 and 5 were remarkable in predicting future disability, which was consistent with previous studies [58,59]. Similar to the traditional method, the assessment of most frailty tools was stratified into three categories (robust, prefrail, frail) [2,31,60]. In the current study, the CFSS-10 would be recommended for assessing subjects as robust (0), pre-frail (1–4), and frail (≥5 points).

There were several limitations that need to be mentioned. First, our participants came from multiple cities which showed geographical advantages; however, the number of subsamples in several cities was inadequate, which likely gave rise to a potential bias that led to a less-than-excellent representative sample of the older Chinese population. In fact, the original plans were disturbed by the sudden SARS-CoV-2 pandemic, and we had to supplement samples from other cities. Second, our research lacked objective measures; in other words, we used the FRAIL scale instead of Fried’s FP according to the FRAIL scale showed a near-equivalent diagnostic ability to FP in a previous study [61]. However, we also believe that it is necessary to compare the new tool with objective measurements of frailty in future studies. Third, the short follow-up duration of this study did not present strong evidence of the predictive ability of the new tool for all-cause mortality. We will conduct further verification studies at different settings and test the predictive ability by following the cohort. Although the multivariable model adjusted for important covariates such as demographic characteristics, there were other potential confounders that may not be investigated in this study, such as BMI and health behaviors. We encourage confounders to be fully considered in future related research. On the positive side, good reliability and criteria validity of the CFSS-10 was presented in this current study, showing it has a certain ability to identify frailty in community-dwelling older adults. Lastly, there was somehow a lack of external criteria validation corresponding to each aspect of ICOPE; however, this study emphasized the development of a valid frailty screening scale. On the other hand, the excellent predictive validity of the CFSS-10 on disability at least provided support for monitoring an individual’s overall intrinsic capacity. Future research may provide evidence for its diagnostic ability on each aspect of IC.

## 7. Conclusions

Although the CFSS-10 did not present perfect reliability and validity tests, it could be used for preliminary screening for frailty among community-dwelling older adults in Shanghai. It worth mentioning that the CFSS-10 provided the six dimensions of the IC assessment that would be useful to support long-term monitoring of functional trajectories to prevent and identify frailty and adverse outcomes. Therefore, we encourage this promising tool to be used for further verification and generalization.

## Figures and Tables

**Figure 1 ijerph-19-11811-f001:**
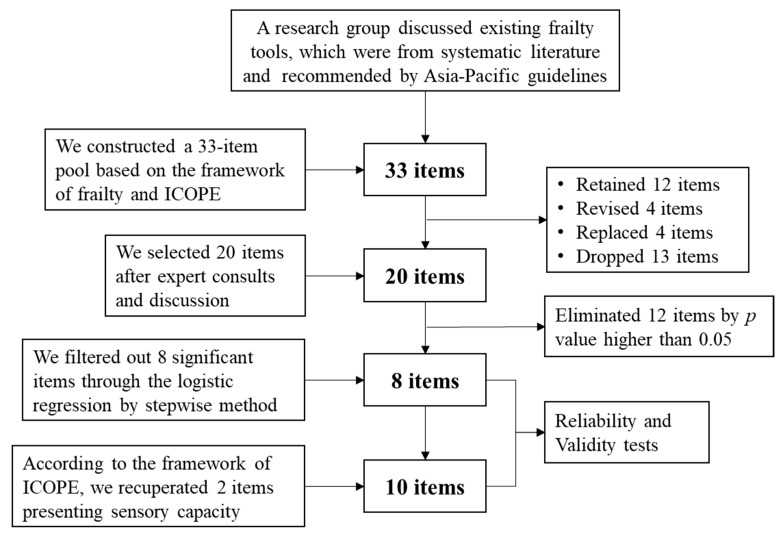
Flow chart of the 10-item Chinese frailty screening scale (CFSS-10) generation.

**Figure 2 ijerph-19-11811-f002:**
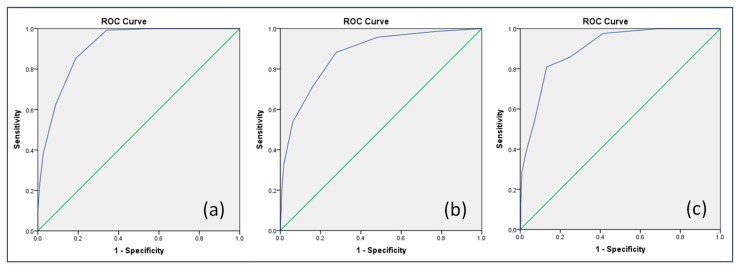
The AUCs of the CFSS-10 taking the FRAIL, TFI and FI as criteria (sample 1). (**a**). AUC of the CFSS-10 = 0.911 (95% CI: 0.894–0.928) taking the FRAIL as a criterion; (**b**). AUC of the CFSS-10 = 0.874 (95% CI: 0.847–0.900) taking the TFI as a criterion; (**c**). AUC of the CFSS-10 = 0.874 (95% CI: 0.826–0.921) taking the FI as a criterion; CFSS: Chinese frailty screening scale; TFI: Tilburg frailty indicator; FI: frailty index; AUC: area under the curve.

**Figure 3 ijerph-19-11811-f003:**
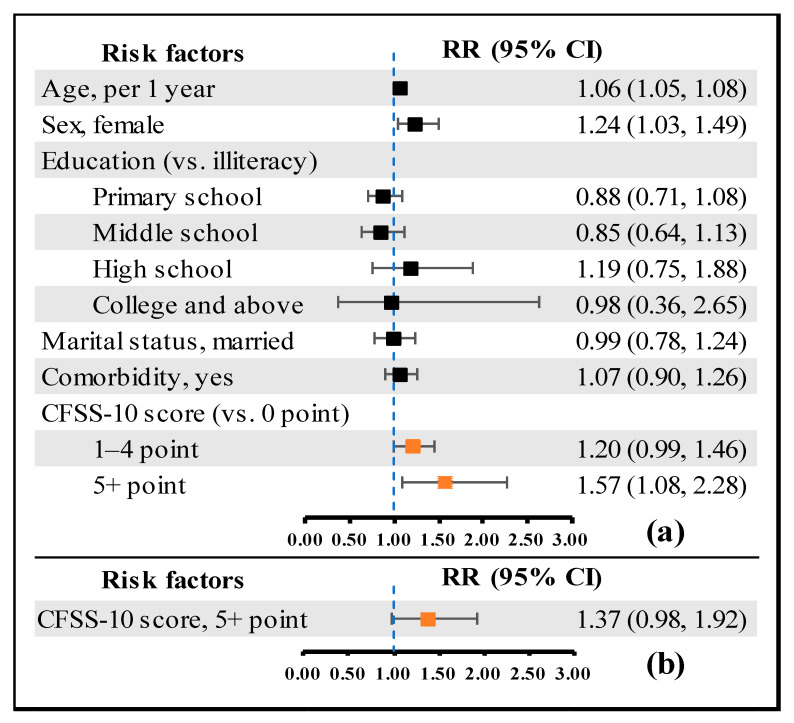
The risk ratios of one-year disability by different cut-off points of the CFSS-10. Model (**a**) and model (**b**) were both adjustment for age, sex, education, marital status and comorbidity. RR: risk ratio; CI: confidence interval; CFSS: Chinese frailty screening scale.

**Table 1 ijerph-19-11811-t001:** Characteristics of sample 1 and sample 2.

Characteristics	(Mean ± SD) or *n* (%)		*p* Value
	Sample 1 (*n* = 1062)	Sample 2 (*n* = 2008)	
Area			
Shanghai	301 (28.3)	2008 (100.0)	
Hangzhou	96 (9.0)		
Nantong	319 (30.0)		
Ya’an	78 (7.3)		
Guangzhou	100 (9.4)		
Jinan	122 (11.5)		
Nanchang	46 (4.3)		
Age, year	76.7 ± 7.2	72.4 ± 6.1	<0.001
Sex, male	499 (47.0)	937 (46.7)	0.864
Education			<0.001
Illiteracy	309 (29.1)	890 (44.6)	
Primary school	307 (28.9)	719 (36.0)	
Middle school	246 (23.2)	321 (16.1)	
High school	108 (10.2)	54 (2.7)	
College and above	91 (8.6)	13 (0.6)	
Marital status			<0.001
Married	806 (76.0)	1678 (84.2)	
Other ^a^	254 (24.0)	316 (15.8)	
BMI, kg/m^2^	24.4 ± 8.7		
GDS score, point	3.0 ± 2.7		
MMSE score, point	20.6 ± 7.3		
Comorbidity, ≥2	541 (72.8) ^b^	831 (41.5)	<0.001
FRAIL score			
1–2 point	586 (55.2)		
3–5 point	252 (23.7)		
TFI score, ≥5 point	237 (31.9) ^b^		
FI, ≥0.25	74 (18.6) ^c^		
ADL, difficulty ≥1 task	212 (20.0)	115 (5.7)	<0.001
IADL, difficulty ≥1 task	493 (46.4)	435 (21.7)	<0.001

SD: standard deviation; BMI: body mass index; GDS: geriatric depression scale; MMSE: mini-mental state examination; TFI: Tilburg frailty indicator; FI: frailty index; (I) ADL: (instrumental) activity of daily living; ^a^ Never married (*n* = 4), widowed (*n* = 243) and divorced (*n* = 7) in sample 1 and never married (*n* = 15), widowed (*n* = 287) and divorced (*n* = 19) in sample 2; ^b^ with sample of 743; ^c^ with sample of 397.

**Table 2 ijerph-19-11811-t002:** The 10-item Chinese frailty screening scale (CFSS-10).

Item	Question	Answer
**Illnesses**	Have you been diagnosed with at least 5 illnesses by doctors? (i.e., Hypertension; Dyslipidemia; Diabetes or high blood sugar; Cancer or malignant tumor (excluding minor skin cancers); Chronic lung diseases; Liver disease; Heart attack, coronary heart disease, angina, congestive heart failure, or other heart problems; Stroke; Kidney disease; Stomach or other digestive disease; Alzheimer’s or Parkinson’s disease; Arthritis or rheumatism; Asthma)	□ Yes □ No
**Exhaustion**	Did you often feel tired or fatigue in the last month?	□ Yes □ No
**Lack of appetite**	In the last three months, did you eat less due to loss of appetite, indigestion, teeth problem or dysphagia?	□ Yes □ No
Visual impairment	Do you experience problems in your daily life due to poor vision?	□ Yes □ No
Hearing loss	Do you experience problems in your daily life due to poor hearing?	□ Yes □ No
**Resistance**	Do you have difficulty with climbing 10 stairs or a flight without resting?	□ Yes □ No
**Physical inactivity**	Did you walk for at least 10 min or 400 m continuously in the last week?	□ Yes □ No
**Attention**	Did you often wander or have difficulty with concentrating in the last month?	□ Yes □ No
**Orientation**	Did you frequently get the date wrong or get lost in the last month?	□ Yes □ No
**Depressive symptom**	Did you feel you were not interested in doing anything in the last month?	□ Yes □ No

Note: the eight bold items of the CFSS-8.

**Table 3 ijerph-19-11811-t003:** Spearman’s correlation between items and total score and the factor analysis of the CFSS-10 (sample 1).

Items	r ^a^	Component			
		Factor 1	Factor 2	Factor 3	Factor 4
Illnesses	0.39 *	0.02	0.75	−0.10	−0.06
Exhaustion	0.59 *	0.35	0.56	0.07	−0.01
Lack of appetite	0.44 *	0.07	0.55	0.17	0.33
Visual impairment	0.49 *	0.18	0.06	0.77	−0.01
Hearing loss	0.45 *	0.02	0.12	0.82	0.01
Resistance	0.51 *	0.07	0.52	0.25	−0.07
Physical inactivity	0.28 *	0.02	−0.03	−0.03	0.93
Attention	0.55 *	0.82	0.02	0.11	−0.02
Orientation	0.53 *	0.78	0.10	0.10	−0.10
Depressive symptom	0.45 *	0.62	0.21	0.03	0.22

CFSS: Chinese frailty screening scale; ^a^ Spearman’s correlation coefficients between items and total score; * *p* < 0.001.

**Table 4 ijerph-19-11811-t004:** Diagnostic accuracy of the CFSS-10 using the FRAIL scale, TFI and FI as criteria (sample 1).

Criteria	AUC (95% CI)	Cut-Off	Youden Index	Sensitivity (%)	Specificity (%)	PPV (%)	NPV (%)	Kappa
FRAIL	0.91 * (0.89, 0.93)	≥3	65.1	99.2	65.9	47.5	99.6	0.47 *
		≥4	66.5	85.3	81.2	58.6	94.7	0.58 *
		≥5	53.5	62.3	91.2	68.9	88.6	0.55 *
TFI	0.87 * (0.85, 0.90)	≥3	55.1	70.9	84.2	67.7	86.1	0.54 *
		≥4	60.5	88.2	72.3	59.9	92.9	0.54 *
		≥5	47.5	53.6	93.9	80.4	81.2	0.52 *
FI	0.87 * (0.83, 0.92)	≥3	58.0	78.4	79.6	46.8	94.1	0.46 *
		≥4	58.0	67.6	90.4	61.7	92.4	0.56 *
		≥5	44.6	48.6	96.0	73.5	89.1	0.51 *

CFSS: Chinese frailty screening scale; TFI: Tilburg frailty indicator; FI: frailty index; AUC: area under the curve; CI: confidence interval; PPV: positive preventive value; NPV: negative preventive value; * *p* < 0.001.

**Table 5 ijerph-19-11811-t005:** The results of the CFSS-10 predicting one-year disability and all-cause mortality (sample 2).

Number of Positive Items	Disability (*n* = 1549)					All-Cause Mortality (*n* = 2008)				
	*n* (%)	RR (95%CI)	*p* Value	Adjusted RR (95%CI) ^a,b^	*p* Value	*n* (%)	RR (95%CI)	*p* Value	Adjusted RR (95%CI) ^a,c^	*p* Value
0	119 (19.0)	reference		reference		3 (0.42)	reference		reference	
1	138 (26.2)	1.38 (1.11, 1.71)	0.004	1.18 (0.95, 1.46)	0.136	9 (1.45)	3.42	0.065	2.49 (0.66, 9.46)	0.179
2	63 (29.6)	1.55 (1.19, 2.02)	0.001	1.28 (0.99, 1.66)	0.062	12 (3.91)	9.24 (2.61, 32.74)	<0.001	5.49 (1.52, 19.92)	0.010
3	26 (25.5)	1.34 (0.93, 1.94)	0.121	1.19 (0.82, 1.72)	0.357	6 (3.59)	8.49 (2.12, 33.95)	0.002	4.42 (1.06, 18.42)	0.041
4	12 (26.7)	1.40 (0.84, 2.33)	0.196	1.13 (0.70, 1.82)	0.628	4 (4.17)	9.85 (2.20, 44.00)	0.003	3.73 (0.79, 17.64)	0.100
5+	16 (43.2)	2.27 (1.52, 3.40)	<0.001	1.57 (1.08, 2.27)	0.017	3 (2.78)	6.56 (1.32, 32.53)	0.021	2.17 (0.40, 11.71)	0.368
*p* for trend		<0.001		0.035			<0.001		0.244	

CFSS: Chinese frailty screening; RR: risk ratio, CI: confidence interval; ^a^ adjusted for age, sex, education, marital status and comorbidity; ^b^ using the log-binomial model; ^c^ using the Poisson model.

## Data Availability

The data applied and analyzed in the current study are available from the corresponding author upon reasonable request.

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
