# Peer review of "Development and Validation of the Chinese Frailty Screening Scale: A Study among Community-Dwelling Older Adults in Shanghai"

_ijerph, 2022, doi:10.3390/ijerph191811811_

Round 1

Reviewer 1 Report

This article discusses the validation of a new tool to assess frailty while taking into consideration the IC. 

The article is interesting and novel. Nevertheless, I have major comments that need to be answered in order to enhance the article:

1- Extensive English revision is needed as there are several grammatical mistakes along the text that affects its readability and understanding

2- The introduction needs to be enhanced. While the idea is there in terms of adding IC to frailty, it is not well presented and the last paragraph of the introduction is a bit confusing since there are frailty indices that take into consideration the physical and mental health that has been validated among the Chinese population. You also mention several validated measures but you do not mention how your suggested index will be novel compared to the previous one. The knowledge gap needs to be better identified. 

3- Figure 1: it is important to justify and present at early stages that you will be choosing specific factors regardless of their statistical significance (i.e you recuperated two sensory factors). How would this affect your models if you kept those as constant from the beginning of the analyses?

4- is n=89 sufficient to test-retest the reliability of the index?

5- The choice of the three frailty scales against which you are validating the simplified score needs to be justified. Why you didn't choose to validate against the frailty phenotype developed by Fried et al (or its Chinese validated version) since it is the most widely used index?

6- Why did the authors choose ADLs as an outcome? it was not justified. This is important to justify as several experts consider that frailty and ADLs are overlapping entities instead of two separate ones. 

7- All-cause mortality is a major outcome but you did not mention how did you measure this. This needs to be added

8- Table 1: Why is BMI missing for sample 2?

9- Table 3: please use lack of appetite instead of "Inappetence"

10- The first question of CFS-10 tackles comorbidity and then when calculating the cut-off point you adjust for comorbidity. Isn't this over-adjustment? what was the rational to include comorbidity in the model?

11- Same comments goes for the results of table 5. 

12- Table 5: the 95%CI needs to be presented along with the RR

13- how the 1-year all-cause mortality might have affected the results? I feel that the a 1-year mortality is too short to be caused by Frailty itself especially that one criterion of the CFS is more than 5 chronic diseases. I believe that this on its own increases risk of mortality. this short mortality risk needs to explained and discussed in terms of how it is affecting the validity of the developed index.

14- the limitations of the study needs to be better discussed: Self-reported nature of data especially for CFS criteria, generalizability of the tool, 1-year mortality risk. 

Author Response

Dear professor,

It is a great honor to receive your comments, which have substantially improved our manuscript. Here provides our responses to your kind advice and concerns, please see the attachment.

Reviewer 2 Report

The work undertaken and the manuscript are interesting and bear value since frailty is one of the most dificult to delineate geriatric giants.

The title states that frailty is about to be tackled from a "healthy aging" perspective in the manuscript. Yet, the concept is not clearly laid out, moreover the authors relate to the WHO definition in one place and that's all. If put into the title, this concept must be stated very clearly. At present, this aspect is not sufficiently taken into account. An easier way would be probably to remove this part of the title.

The literature review is comprehensive, the statistics are robust, the conclusions seem a bit too categorical, given the kappa value of a bit over 0.6.

The description of sampling lacks precision. In general, despite the commonly accepted sample calculators, the sample seems small, given the size of China in terms of country and population, particularly when the authors use the adjective "nationwide". At least, this aspect should be discussed.

The manuscript deserves a moderate English overhaul. Moreover, the use of the world "elderly" is considered inappropriate as it may be perceived as offensive. One may use "older people" or "older adults" instead.

Author Response

(The authors gave the same response as above.)

Reviewer 3 Report

Development and validation of the Chinese Frailty Screening scale from healthy aging perspective: a study among Community-dwelling older adults in Shanghai(ijerph-1857008)

 This manuscript aims at developing and validating of the Chinese Frailty Screening scale to effectively screen frailty effectively. Overall, this topic is important and the methodology is sound. Only a minor concern appeared after reading the whole manuscript.

 The recent paper also explored the similar topic and should be reviewed and discussed.

 Egashira, R., Sato, T., Miyake, A., Takeuchi, M., Nakano, M., Saito, H., ... & Hagihara, K. (2022). The Japan Frailty Scale is a promising screening test for frailty and pre-frailty in Japanese elderly people. Gene, 146775.

Author Response

(The authors gave the same response as above.)

Reviewer 4 Report

1.  This manuscript adds to the development of the concept of frailty and utilizes the World Health Organization concept of intrinsic capacity.

2.  The methodology appears sound.  Some suggestions regarding the structure of the report are provided for the authors' consideration.

3.  Some modification of language will improve the clarity of the manuscript.  Some examples are provided in the abstract, introduction, and conclusion.

Abstract: [delete words in brackets]

Background: Based on intrinsic capacity (IC) defined by World Health Organi-zation, an accelerated downhill trajectory of the IC may present an important precursor for older adults occurring frailty. Considering the lack of validated instruments, which combined screening frailty and monitoring IC, this [. This] study aims to develop a comprehen-sive and acculturative frailty screening scale among Chinese older adults from a perspec-tive of healthy aging. Setting and participants: A cross-sectional study and a cohort study both based on community-dwelling older adults aged 65 years old and above. Methods: This study [mainly] consisted of three [two parts]. First[ly], we selected and revised 20-items related to frailty based on literature review, expert consultation and stakeholder analysis. Sec-ond[ly], a nationwide cross-sectional study was conducted to further simplify the scale, and to test the reliability and validity of the new frailty screening tool. The FRAIL scale, the Tilburg frailty indictor (TFI), and a 49-item Frailty Index (FI) were investigated as [the] cri-teria. Third [Additionally], a cohort study in Shanghai was implemented to verify the predictive validity of the new frailty screening scale. The disability measured by the activity of daily living (ADL) and instrumental activity of daily living (IADL) and all-cause mortality were documented as outcomes. Results: A 10-item scale named Chinese frailty screening scale (CFS-10) was successfully developed and validated. It presented the Cronbach’s α of 0.63 and the intraclass correlation coefficient of 0.73, indicating good reliability. Taking the other frailty tools as criteria, the Kappa values of 0.54~0.58 and the area under the curves of 0.87~0.91 showed good validity of the CFS-10. The results of the Log-binomial models and the Poisson models showed a high[er] score of the CFS-10 predicting a higher risk of disability and all-cause mortality. An optimal cut-off point[s] of 5 has excellent prediction of 1-year disability. Conclusions: the CFS-10 has good validity and reliability as a quick and acculturative frailty screening tool for older adults in Shanghai. It may complement other existing  [be asupplementary for the existed] frailty screening tools among Community-dwelling older adults. [This new tool is worthy to be further verified and promoted nationwide in the future.]

Keywords: frailty; assessment tool; disability; older adults; healthy aging

Introduction

Frailty is [was] defined as a progressive decline of physiological reserves conferring in-creased older adults’ vulnerability to stressors and exposing to higher vulnerability of ad-verse health outcomes, including disability, falls, hospital admissions, and mortality (1, 2). A meta-analysis showed the prevalence of frailty among adults aged ≥50 was 12%~24% worldwidely (3). The prevalence of frailty was from 6% to 25% across peoples in 65-74 years to those ≥85 years with an average of 10% in China (4). However, the global or re-gional prevalence of frailty is not yet accurately known, and is [as] an important reason of the different operational definitions of frailty across studies. Even so, frailty represents a pub-lic health priority for its highly and increasingly prevalent condition in the aging popula-tion worldwidely (5).

To cope with the increasing challenges of aging population, the World Health Or-ganization (WHO) proposed a framework of healthy aging (6). The WHO also emphasizes that it is important to develop an approach to screen individual’s decreasing in intrinsic capacity (IC). The IC is defined as “the combination of the individual’s physical and men-tal capacities” (7). It was developed with taking as background the evidence built on frailty over decades and to nest into the framework of healthy aging (8). The “Integrated Care for Older People” (ICOPE) was developed as a guidance to screen for loss of IC, with 6 areas including cognitive decline, mobility, malnutrition, vision impairment, hearing loss and depressive symptoms (9). Except for cognitive and sensory capacity, the other capacities overlap [are overlapped] with frailty phenotypes. The IC indicators, such as gait speed, grip strength and exhaustion, were also used in frailty definitions, while the cognition tests were also used for screening cognitive frailty (10). Recently, the ICOPE tool was also recommended to screen frailty by the International Conference of Frailty and Sarcopenia Research (ICFSR) consensus guidelines (11). The IC could be used throughout the life course and regarded as a comprehensive aspect of individual’s functional capabilities, while frailty was regarded as an accelerated downhill trajectory before disability occurs in the latter part of the life course. To a large extent, decreasing in the IC probably be an important precursor cause for older persons occurring either prefrailty or frailty (10, 11). According to both existing assessment frameworks, an abnormal decline of an individu-al's IC in older age may coincide with the onset of frailty. Using a life-course approach might increase our understanding of how frailty and its risk factors develop in earlier life stages, and could contribute to the development of public health strategies for frailty pre-vention (12).

In the past decade[s], numerous tools developed to screen or assess frailty have been published (13). However, there is still no global standard assessment instrument for frailty (12). The most widely used measures are the Frailty Phenotype (FP) (2) and the Frailty Index (FI) (14), but they are not easily implemented in [large] population research[es] or busy clinical practices, as the former requires objective measurements and the latter needs to collect too much information (13). In recent years, a Chinese self-reported frailty screening questionnaire (FSQ) was developed based on modified Fried FP criteria (15), and vali-dated in different settings (16). Besides, previous studies have developed frailty scales based on the FI for the elderly in China (17, 18). But one of them, the FI-35 with an oversize of 11 dimensions, might produce a potential third-order model to some extent (19). More-over, the FI-35 was not easy to screen frailty in the large-scale population, as the samedisadvantage with other instruments based on the FI (13). Subsequently, several simple and effective screening questionnaires emerged. The FRAIL scale is a very simple tool for screening frailty and was previously used among Chinese older populations (20, 21). However, the FRAIL scale has its weakness, such as eliminating the low physical activity may change the theoretical model of the FP (22). Increasingly, frailty is considered a mul-tidimensional concept, mainly including the physical, mental (psychological and cogni-tive) and social frailty. A Chinese version of the Tilburg frailty indictor (TFI) was also developed as an integral instrument to measure frailty among community-dwelling older adults in China (23). Whereas, the social frailty was similar with the social isolation, which was the external cause for frailty (24). Physical frailty and cognitive frailty are more widely accepted, which was probably the reason why the WHO compared IC with frailty (25) and why the researchers regarded them as two related concepts (8, 10).

[As saying above, m]Monitoring the IC in the early stage of aging is benefit for identify-ing the process of frailty. Therefore, this study aims to develop and validate a Chinese frailty screening scale based on utilizing the WHO IC framework. [the definition of frailty (physical and cognitive aspect), at the meanwhile considering the framework of IC, to rapidly detect IC and screen frailty in older adults.]

.

.

Part two: revising and validating the CFS

Firstly, a nationwide cross-sectional study among community-dwelling older adults was conducted to simplify the items of the CFS-20. Multi-stage random sampling was used to select subjects. Five cities were purposefully selected in five geographic areas in China (east, west, south, north and center), respectively. And then two communities were randomly selected from urban and suburban/village in each city. Simple random sam-pling was used to select older adults lived in community. Actually, we added samples from other cities due to the influence of the COVID-19 pandemic. A total of 1083 partici-pants aged 65 years and above were originally recruited. After eliminating participants with missing data, the valid response rate was 98.1% (1062/1083) (sample 1). To determine the test-retest reliability of the simplified frailty screening scale, 89 participants were as-sessed twice using the simplified frailty screening scale within a 2-week interval.

In this cross-sectional study, the sample size was calculated by the formula: n=Z2×(1-P)/(P×ε2), the indicator of Z is the percentile corresponding to an area of 0.975 (1-0.05/2) under the standard normal distribution, as the prevalence of frailty was 10%~15% in Chi-nese community-dwelling older adults, set the indicators of P to 12.5% and ε to 0.2, the minimum sample size needs 673. An additional 20% of the sample size would be added to in consideration of the response rate. Therefore, it is better to enroll more than 808 par-ticipants for the current study. The participants were invited to participate in the study and received face-to-face interviews to complete the questionnaire by trained community health workers. Those unable to communicate effectively or with severe cognitive impair-ment were excluded because of the self-reporting nature of data.

[Consider: Part Three: Validating the CFS]  Secondly, a cohort study based on community-dwelling older adults in Shanghai was implemented to verify the predictive validity of the simplified frailty screening scale.

.

.

Conclusion [delete words in brackets]

In summary, the CFS-10 is a reliable and efficient tool that can be used clinically to screen  [as a prelimi-nary screening] for frailty among older adults in Shanghai. It [provides a perspective of healthy aging, which would] may be useful  [be beneficial] to support long-term monitoring of functional trajecto-ries to identify and prevent frailty and other adverse outcomes in older populations. [Thus, this new tool is worthy to be further verified and promoted nationwide in the future.]

Author Response

(The authors gave the same response as above.)

Round 2

Reviewer 1 Report

I acknowledge the efforts that the authors did and I thank them because they addressed all my comments.

Nevertheless, I still have some concerns.

Introduction:

"the frailty index model and the multidimension frailty model": What are these models? this is the first time you mention them without introducing them to the reader.

"However, its weakness is that it eliminates low physical activity, which may change the theoretical model of the FP [22] and the same as the PF that is lack of mental aspect" what is PF? and what do you mean by the newly added sentence?

Methodology:

To which extent the information regarding mortality gathered from community workers is reliable?

The choice of the three frailty scales against which you are validating the simplified score needs to be justified in the manuscript. Also, are those scores subjective or they have objective measures? 

The authors should mention that BMI values were not available for sample 2. 

I am still not convinced of the adjustment of comorbidity in the analysis. What is the correlation between Illness and comorbidity variables? Please do a sensitivity analysis where you remove this variable from the adjustment variables and see how this would change the results. 

 Discussion:

"In the other hand, those with higher score of the CFS-10 at baseline might get more attention and health care". It can also mean the duration is too short to observed an actual effect on mortality. and this needs to be discussed. 

There are still some English grammar mistakes:

"During the next year, there 141 participants were lost to follow-up and 37 died" You either put there were 141 participants or you remove there.

"with the death information obtaining" should be " obtained" instead. 

"Similarly, in recent, researchers from non-native English-speaking" Replace "Similarly, in recent," by "more recently"

"so that it did not observe the outcomes in a short period of follow-up" "it" refers to who?

"Although the CFSS-10 presented not at all perfect in reliability and validity tests," this sentence needs to be corrected.

Author Response

Dear professor,

It is a great honor to receive your comments, which have substantially improved our manuscript. Here provide our responses to your kind advice and concerns. Please see the attachment
